# Approaches to enabling rapid evaluation of innovations in health and social care: a scoping review of evidence from high-income countries

Gill Norman ![ORCID],[1,2] Thomas Mason ![ORCID],[3,4] Jo C Dumville ![ORCID],[1,2] Peter Bower ![ORCID],[2,3] Paul Wilson ![ORCID],[2,3] Nicky Cullum ![ORCID] [1,2]

For numbered affiliations see end of article.

**Correspondence to**
Dr Gill Norman;
gill.norman@manchester.ac.uk

## ABSTRACT

**Objective** The COVID-19 pandemic increased the demand for rapid evaluation of innovation in health and social care. Assessment of rapid methodologies is lacking although challenges in ensuring rigour and effective use of resources are known. We mapped reports of rapid evaluations of health and social care innovations, categorised different approaches to rapid evaluation, explored comparative benefits of rapid evaluation, and identified knowledge gaps.

**Design** Scoping review.

**Data sources** MEDLINE, EMBASE and Health Management Information Consortium (HMIC) databases were searched through 13 September 2022.

**Eligibility criteria for selecting studies** We included publications reporting primary research or methods for rapid evaluation of interventions or services in health and social care in high-income countries.

**Data extraction and synthesis** Two reviewers developed and piloted a data extraction form. One reviewer extracted data, a second reviewer checked 10% of the studies; disagreements and uncertainty were resolved through consensus. We used narrative synthesis to map different approaches to conducting rapid evaluation.

**Results** We identified 16 759 records and included 162 which met inclusion criteria.

We identified four main approaches for rapid evaluation: (1) Using methodology designed specifically for rapid evaluation; (2) Increasing rapidity by doing less or using less time-intensive methodology; (3) Using alternative technologies and/or data to increase speed of existing evaluation method; (4) Adapting part of non-rapid evaluation.

The COVID-19 pandemic resulted in an increase in publications and some limited changes in identified methods. We found little research comparing rapid and non-rapid evaluation.

**Conclusions** We found a lack of clarity about what 'rapid evaluation' means but identified some useful preliminary categories. There is a need for clarity and consistency about what constitutes rapid evaluation; consistent terminology in reporting evaluations as rapid; development of specific methodologies for making evaluation more rapid; and assessment of advantages and disadvantages of rapid methodology in terms of rigour, cost and impact.

## STRENGTHS AND LIMITATIONS OF THIS STUDY

⇒ Strengths of this study include a prespecified, published review protocol which anticipated the need for flexibility in the structure and focus of a wide-ranging scoping review. Our paper adheres to internationally adopted recommendations for reporting scoping reviews (Preferred Reporting Items for Systematic Review and Meta-Analysis for scoping reviews (PRISMA-ScR)).

⇒ A further strength is our search strategy which was developed and implemented with an information specialist and covered multiple databases.

⇒ One limitation of this study is our dependence on whether and how study authors described their studies as 'rapid' and the lack of clarity in the reporting of the included studies.

⇒ A further limitation is that we did not screen or chart every study in duplicate, however analysis of those that we did assess in duplicate suggests this is unlikely to have impacted substantially on the review content.

## INTRODUCTION
### Background

The need to support innovation in health and social care sits alongside the need to ensure that innovations are appropriately evaluated to ensure safe, effective and cost-effective care for patients and service users. In the absence of adequate evaluation new technologies or practices may be adopted which do not have proven benefits and which may even result in harm. However, evaluation needs to be conducted quickly and efficiently if it is to provide timely information for decision-makers. There is increasing demand for timely, rigorous evaluation of innovation in health and social care. Demand has been amplified during the COVID-19 pandemic, where the ability of research infrastructure to respond rapidly has been paramount.

The term 'rapid evaluation' is widely used to describe approaches that aim to adopt pragmatic methods for timely assessment of innovations, but there is uncertainty about what this means in practice. Rapid evaluation can be conceptualised to include rapid conception and inception of evaluation in response to an identified need, or rapid completion of part or all of the evaluation process, but linking these ideas to practice still requires investigation.

While the need for rapidity in evaluation is widely recognised, there has been limited assessment of how rapidity can be actually achieved[1–3]; this is particularly the case for primary evaluations as opposed to rapid reviews.[4–8] There is a growing body of research on the development of formal methods specifically designed for rapid evaluation,[9–11] but this forms only part of a wider literature reporting 'rapid' studies.

Scoping reviews of evaluation approaches can map available approaches, identify gaps where further development is needed and provide insights into the strengths and weaknesses of existing methods.[12 13] Understanding the different types of approaches and the range of methods adopted by researchers to achieve rapidity of evaluation in practice may inform the development of new approaches to evaluation of innovations. It also has the potential to provide additional insights into the trade-offs and limitations which may result from rapidity in terms of rigour or transferability of findings.[14] Finally, it may identify research questions in which there has been limited use of rapid evaluation approaches and where further development might be prioritised.

We therefore undertook a scoping review to map reports of rapid evaluations of health and social care innovations; map the types of question asked in rapid evaluations and to identify where further research is needed to support more rapid evaluation. We focused the review on studies from high-income countries to increase the relevance to the National Health Services in the UK.

### Research questions

In this scoping review we explored the following research questions:

► What methodologies and methods relevant to a UK context have been used to undertake rapid evaluations of health and social care innovations? How are these approaches expected to deliver rapidity?

► What is the comparative evidence for rapid evaluation methodologies? Have these methodologies been compared with non-rapid standard (however defined) methods or other rapid methodologies?

### METHODS

#### Approaches to scoping for rapid evaluations

In this scoping review, we aimed to: identify and summarise evaluations addressing health-related questions in ways which had been described as 'rapid'; document the ways in which study authors considered that evaluation was made 'rapid'; and summarise evidence that compares the risks and benefits of rapid evaluation and alternative approaches. We accepted authors' definitions of what constituted 'rapid evaluation'. We considered evaluations of the safety and effectiveness of innovations as well as a wider set of outcomes, including acceptability and user experience. In this scoping review we did not seek to assess the quality of studies.

We adopt the definition of innovation as: *any novel technology, device, procedure, set of behaviours, routine or way(s) of working that is directed at improving health outcomes, administrative efficiency, cost-effectiveness or users' experience—and is implemented by planned and coordinated actions.*[15] Thus, innovation can include alterations in service delivery, organisation and financing, as well as clinical and care practices.

We considered evaluation as a complex and multifaceted undertaking which lacks a single definition. Using a broad definition that encompasses research, typically evaluation investigates whether an overall net benefit is gained from innovation over alternative activities. Phase III randomised controlled trials (RCTs) and systematic reviews of innovations remain the 'gold-standard' of comparative effectiveness methods; although quasi-randomised approaches are often more feasible. For other uncertainties around innovations (eg, acceptability, feasibility) there are well established quantitative, qualitative and mixed methods (and corresponding approaches to evidence synthesis).

We adopted an iterative approach to review methods which we planned for in our protocol, which was completed before the screening stage of the review and published on OSF (Open Science Framework).[16] In particular, we have narrowed the scope of the review and focused on addressing a subset of the original questions posed. We have done this in order to engage fully with the questions we considered most central to our objectives (see above). We have followed the reporting approach recommended in the extension of Preferred Reporting Items for Systematic Review and Meta-Analysis for scoping reviews where possible given this planned flexibility.[17]

### Inclusion eligibility

Our inclusion criteria (box 1) were developed using a PICOS (Population, Intervention, Comparator, Outcome, Study type) approach and used to screen the results of the database search. We developed the inclusion criteria iteratively to ensure that the most relevant research was captured, anticipating this flexibility in the protocol.

We only included publications in English. We planned to document relevant non-English publications but not to extract data or include them due to the available resources. However, the great majority of the non-English publications identified were ineligible (based on assessment of an English language abstract).

### Search

We searched PubMed/MEDLINE, EMBASE and Health Management Information Consortium (HMIC) databases. This main search was updated on 13 September

## Box 1 Eligibility criteria

**Study design:** Publications which evaluate, compare, document the use of, provide guidance on or otherwise describe one or more rapid evaluation methods for assessing innovations according to the study authors' definition (of rapid evaluation). We also included previous systematic or scoping reviews of rapid evaluations but noted overlap with included primary studies. We did not include rapid reviews or papers relating to methods for rapid reviews. Where studies used a rapid review as the first stage of primary evaluation, we included them only if the primary evaluation element was also described as rapid or using rapid methods. Evaluations could be primarily quantitative or qualitative in focus (or use mixed methods). We also excluded pilot studies which were not explicitly described as rapid evaluations.

**Innovations (interventions):** Publications on rapid evaluation of innovations or their implementation in the field of health and of social care as it relates to health. Innovation is broadly defined as encompassing intentional change.[15] We subsequently broadened this to encompass interventions or services not described as innovations. We excluded publications reporting methods which related only tangentially to interventions or services such as Rapid Assessments of Needs and Rapid Epidemiological Assessments.

**Populations:** We included any study relating to rapid evaluation methods and did not further limit eligibility based on the study population beyond the requirement that the study be undertaken in a health or related care setting in a high-income country using World Bank definitions.[236] We also excluded studies which looked at evaluation in a regulatory context because these represented accelerated evidence review processes and were therefore considered to represent review methodology rather than primary evaluation.

**Outcomes:** We accepted any health-related outcome measure; this includes any clinical measure relevant to the intervention including patient-reported outcomes or experience; measures of care process or service use; and measurements of costs or activity as well as health economic measures such as cost-effectiveness. This was used to ensure the relevance of the study to health and related social care.

2022, having initially been conducted in February 2020. We also looked at websites of key organisations involved in rapid evaluations (eg, RSET, BRACE, Nuffield Trust and Health Foundation, THIS Institute) and checked references of identified studies. Finally, we searched Google Scholar and checked the first 1000 records against the main database searches. The database search strategy (online supplemental appendix 1) was designed and implemented by an information specialist. We selected terms to balance sensitivity and specificity. There were two main elements to the initial search: a search for rapid evaluations undertaken from database inception, and a search from 2014 onwards using the search strategy from a previous, high-quality scoping review of rapid reviews which had searched up to 2014.[5]

We subsequently narrowed the scope of our search to exclude rapid evidence synthesis (ie, rapid reviews (box 1)) as it became clear that the volume of recent rapid reviews and related methodological papers would require a separate piece of work. Given that considerable work on rapid review methodology is being undertaken,[18–21] we therefore took the decision to focus on primary research.

Our update search in September 2022 did not include this facet. While we excluded papers on rapid synthesis, we nevertheless included syntheses of rapid primary studies (box 1).

### Screening

One researcher screened initial search results by title and abstract in EndNote and selected potentially relevant records for full-text evaluation.[22] A 5% sample was exported to Microsoft Excel (V.365) and screened by a second researcher for consistency. Records obtained at full text were screened by one researcher, with a second researcher assessing 10% to check consistency, and subsequently available for discussion in cases of uncertainty. We did not undertake duplicate screening for all records because of the high volume of retrieved records and because the sample which were duplicate screened showed a very high level of consistency (at abstract stage we identified one additional record requiring full-text assessment for every 200 records dual screened; only a single included study was identified which would have been excluded in error). We adopted a conservative approach to full-text exclusion decisions and resolved uncertainty through discussion.

### Data extraction

We extracted sufficient information to enable us to classify studies on their key characteristics and to thereby 'chart' the available literature.[17 23] Following the recommendations of Levac[24] we piloted a bespoke data form using Microsoft Excel to enable data extraction, using two researchers, with a small sample of included studies. The following data were extracted by one researcher (with a

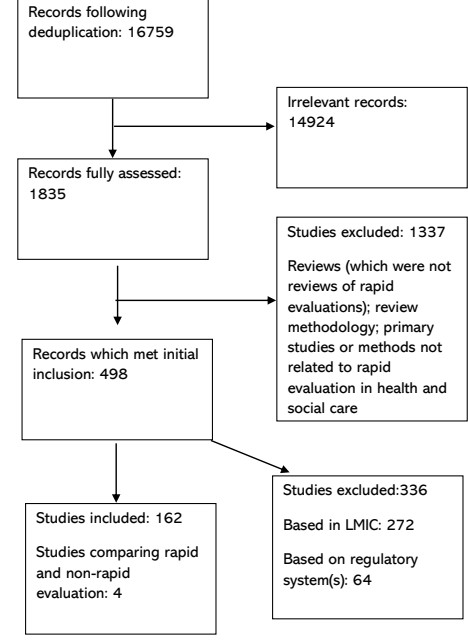

**Figure 1** Preferred Reporting Items for Systematic Review and Meta-Analysis diagram showing study flow for the review. LMIC; low-income and middle-income country.

10% sample cross-checked by a second researcher, who was also consulted in cases of uncertainty).

► Study ID (first author, year of publication).
► Country (individual); country group (Africa, Asia/Middle East, Europe, North America, Central/South America), OECD (Organisation for Economic Cooperation and Development) class (low and middle income (LMIC) or high income).
► Definition(s) of rapid evaluation used by authors including timescale and whether the focus was evaluation/implementation/methods.
► Study designs as described by authors (qualitative/quantitative/mixed methods).
► Question(s) the evaluation sought to address.
► Collection/analysis methods (eg, survey, observation, interviews).
► Nature of innovations evaluated (if practical example).
► Impact of rapid methods (for comparative studies).
► Stage, scope, characteristics and purpose of evaluation for example, local/regional/national; formative/iterative/summative.
► Evaluation methods adaptation to context/complexity of intervention including whether stakeholder participation was reported. This was based on the relevant items from the TIDieR (Template for Intervention Description and Replication) checklist.[25]

We undertook this charting across all studies that met our inclusion criteria. These data were then used to map studies in terms of what made the study 'rapid'. The classification was undertaken by one researcher and discussed by the review team.

### Synthesis and analysis

We developed a narrative synthesis of the evidence relating to each of our two review questions. We structured the synthesis by which question the evidence was addressing and by the methods used to make evaluation rapid. We used data charting to identify studies which shared common features and to group them into categories based on the different approaches to enabling rapidity which were represented. We used discussion and consensus to refine these categories. The synthesis was supported by structured tables to inform the approaches identified and the ways in which these were implemented.

## RESULTS

### Results of the search and screening process

We identified 16 759 unique records (figure 1). Preliminary assessment determined 1835 records to be potentially eligible from which we excluded 1337 records of rapid reviews, primary studies not related to rapid evaluation or studies not reported in English. We excluded a further 272 records of studies in LMICs and 64 which looked at regulatory process evaluations (a list of excluded studies is available on request).

We included 162 studies in the review (table 1); these were methods papers, primary evaluations or evaluations of implementation undertaken in high-income countries; one was a systematic review[9] which included eight of the primary studies included here.[26–32] Ninety of the included studies were identified from the update search in September 2022. Most studies were undertaken in primary care, community settings or in specific non-hospital-based services (often described as 'clinics'), reflecting the health and related care focus of the review. Further details of the included studies are reported in Appendix 2 in online supplemental appendix 2 tables 1–4.

### Methodologies and methods used for rapid evaluation

After mapping included studies, we identified four categories that summarise approaches employed with the aim of achieving rapid evaluation (table 1):

1. Use of a methodology designed specifically for rapid evaluation which involved the whole research process.

| Table 1 | Summary table of included studies | |
|---|---|
| **Means of making evaluation rapid\*†** | **Examples of approach** |
| Methods specifically designed for rapid evaluation: 117 studies with supporting methods papers. | Rapid evaluation or assessment/appraisal process/procedure and adaptations.[9 14 26 33–83 137 237 238] Rapid assessment and response and adaptations.[11 38 39 44 84–91 239] Rapid participatory appraisal and adaptations.[48 92–98] Rapid ethnographic assessment/ethnography and adaptations.[99–122 139 240] Methods from quality improvement.[27 28 30–32 98 123–132 241 242] |
| Doing less or using a less time-intensive version of an existing methodology: 28 studies. | Shorter research period.[140–143] Less intensive method.[58 83 138 144–154] Fewer participants or fewer/smaller study sites.[63 138 139 155–162] |
| Adaptation of part of a standard method: 44 studies. | Priority setting.[133 163 193 194] Ethics approval, recruitment and participation, barriers.[27 145–150 154 164–173 195 196] Study design.[174–180 243] Data collection and analysis.[151 153 175 181–192] |
| Use of alternative technology or data source: 28 studies. | Automation or otherwise increased speed of data collection, collation or analysis.[58 138 145–151 153 154 197–199] Acquisition or use of existing data sets.[129 173 181 200–210] |

\*Some studies appear in more than one category because multiple means were employed.
†Some studies in each category are represented by multiple references.

2. Increasing rapidity by doing less, or using a less time-intensive methodology.

3. Adaptation of a non-rapid evaluation to make this a rapid part of the wider study (eg, the recruitment stage of a trial or the analysis phase of a qualitative study).

4. Use of alternative technologies and/or data sources to increase rapidity of an existing evaluation methodology.

Some studies used more than one approach to rapidity; (eg, an adaptation of a part of a wider study combined with the use of an alternative technology) and are therefore represented in multiple categories.

Below we expand on each category, giving illustrative examples for each.

### Methodology specifically designed for rapid evaluation (rapid methodology)

The largest number of papers concerned methodological approaches used specifically for rapid evaluation (examples and methodologies detailed in Appendix 2, online supplemental appendix 2 table 1). These were represented by 117 studies.[9–11 26 28 30–136] Some studies are represented by more than one publication and by supporting methods papers identified from references. Individual study details are in online supplemental appendix 2 tables 2–4.

Examples include methods described as rapid assessments or rapid assessment studies,[77–82 135 137] rapid assessment procedures or processes;[38 39] rapid evaluation;[55 77–82 135] rapid appraisal procedures or processes;[51 53 59] and rapid ethnographic assessments, ethnographic appraisals or ethnography.[99–103 108–113]

Many of these approaches were closely related while some were explicit adaptations of other rapid approaches, such as rapid appraisal,[59] rapid participatory appraisal[48 96] rapid participatory appraisal with meta-planning,[95] qualitative rapid appraisal, rigorous analysis;[57] rapid assessment and response,[44] rapid assessment response and evaluation,[87] focused rapid assessment process[73] and rapid assessment procedure informed clinical (or community) ethnography.[104 105 117 118] A paper based on Rapid Assessment of Avoidable Blindness, an approach specific to vision services, was also identified,[72] although most examples of this were excluded because they related to LMIC.

Four studies described themselves as adaptations of rapid assessment processes for specific evaluation purposes.[33 34 37 40] Some approaches, such as 'plan, do, study, act', rapid cycle evaluation and rapid feedback evaluation were drawn from areas such as quality improvement and wider implementation science.[30–32 98 123 124 126 130–132]

Four of the papers examined limitations of a particular methodology, ways in which these limitations could be reduced,[115] or looked at the impact of rapid applications of these methodologies.[46 106 107] This included highlighting the need for researchers to have received an appropriate level of training in the methodology.

The overarching characteristics of the methodologies designed to increase rapidity were that they were qualitative or mixed methods. None were exclusively quantitative nor were capable of *comparative* assessment of innovations (as is often required for health technology assessment). The actual methods employed in these studies shared core features including: multiple assessment methods (eg, a combination of interviews of a small number of participants and wider surveys), and triangulation of data from the different approaches, either different methods (eg, surveys and interviews) or the same approach with different groups (eg, interviews with different types of stakeholders). The types of evaluation questions addressed in these studies largely focused on the applicability of previous research; acceptability of interventions; barriers and facilitators to intervention implementation or access; and stakeholder views and experiences.

In addition to the marked increase in the number of studies identified, the principal change we identified in those conducted during the COVID-19 pandemic was the use of online recruitment and videoconferencing software within established methodologies.

### Increasing rapidity by doing less, or using a less time-intensive version of an existing methodology

The second approach to rapid evaluation involved undertaking evaluation on a smaller scale or in a less-intensive way (table 2); sometimes with overlap with use of an alternative technology (see section 4 below).[58 138] We identified 28 studies or related methods papers of this type[58 63 83 138–162] (see online supplemental appendix 2 tables 2 and 3 for details).

Sometimes authors carried out a smaller study, with fewer sites and/or participants meaning that less data collection and analysis was required. Shortening the data-collection period was also adopted.[140–142] One of these studies[143] comprised one of the few comparative studies identified in the review (see section Studies comparing a rapid with a non-rapid method, below). Some studies balanced the impact of using smaller samples by employing techniques identified in methods designed specifically for rapid evaluation above, such as stating that they used random or pseudo-random participant/site selection, or selected a subsample of participants in such a way to ensure representativeness.[138 158 161] In other cases, no such measures were reported; such studies accepted—explicitly or implicitly—the trade-off between rapidity and representativeness or reliability of research.[159] An alternative approach involved using a less-intensive method resulting in reduced information collected for each participant or site. This included using a briefer form of a research tool (survey, questionnaire or interview proforma).[63 138 144 160] A variation on this was the use of group interviews as a substitute for individual interviews.[152] A proposal for an RCT using this principle of doing less at every stage was also identified.[162] Updating the search in 2022 identified for the first time studies which used the internet to make their research less time intensive. Examples of this identified as making a study rapid were online surveys, online recruitment or videoconferencing software for interviews or focus groups.[145–151 153 154] These are primarily examples

**Table 2** Approaches to increasing rapidity through less evaluation/less time-intensive evaluation

| Approach employed with aim of increasing rapidity | Methods | Example |
|---|---|---|
| Shortening response period | Survey; safety monitoring cohort. Survey also used increased number of reminders to improve response rate. | Joseph and Goddard, Demeulemeester et al, Banga et al[140–142] |
| Using an online survey | Survey hosted on webpage and/or social media sites. | Attal et al, Daviskiba et al, Elsawy et al, Hammond et al; Khubchandani et al, Talati et al, Petrella et al[145–150 154] |
| Using telephone or videoconference instead of face-to-face interviews | Interviews. Additional measures included purposive sampling following identification on social media. | Williams et al, Wichmann et al, Fisher et al, Srinivasan et al[58 138 151 152] |
| | Focus groups. Use of breakout rooms within a larger meeting on videoconference platform. Group interviews instead. | Keniston et al[153] |
| Using a rapid interview method | Interviews; need to apply rapid method appropriately noted. Using a group interview instead of individual interviews. Use of rapid data collection methods. | Hamilton, Srinivasan et al, Wanat et al[83 144 152] |
| Using fewer participants in interviews | Interviews. Additional measures included random or quasi-random sampling of participants; use of samples selected for representativeness; use of small core group for all stages of the project. | Wichmann et al, Dunbar et al, Brown-Johnson et al, Hamarman et al, Quintero Romero et al, Loosier et al[138 139 155–158] |
| Using fewer study sites | Random or quasi-random site selection (eg, housing blocks); in some instances the selection of sites/participants is the same process. | Brown-Johnson et al, Hamarman et al, Cakir et al, Rodriguez et al[156 157 160 161] |
| Evaluating a small/low volume site | Site audit/evaluation—acknowledged small site volume made rapidity possible. | Delaney et al[159] |
| Using a shorter timescale for pre/post data periods | Interrupted time series—non-rapid application of method also used (more pre and post data). | McCarthy et al[143] |
| Using simpler methods and recording less data at fewer time points | Asking fewer questions at each time point in a single or repeated measures survey. Use of fewer sites and convenience sampling also reported. Proposal for 'Large Simple Randomised Trials' (wide, simple eligibility criteria; central randomisation; recording few baseline data; reduced/no follow-up visits; outcome assessments of hard endpoints, preferably by registries). This is proposed for simple short-term treatment for large populations in relation to universal vaccinations. | McDonald et al, Rodriguez et al, Hasford[63 160 162] |

of using technology to make research rapid but are included here due to the time-intensity implications of their use.

### Adaptation of part of a standard evaluation method

Forty-four studies reported methods they used with the aim of increasing the rapidity of a discrete element or stage of an evaluation, where the evaluation as a whole used standard methods (table 3)[133 145–151 153 154 163–192] (see online supplemental appendix 2 tables 2 and 3 for details).

This approach often targeted the earlier stages of an evaluation, including research priority setting,[133 163 193 194] ethical approval[164] and participant identification and recruitment.[165–167 195 196] Increased use of online recruitment and data collection was identified following the onset of the COVID-19 pandemic.[145–150] Developments in RCT design were also seen over this period.[173 174 176–179]

At later stages of the evaluation process we identified continuous or contemporaneous data collection or data analyses as a means to increase rapidity of evaluation in RCTs and other quantitative study designs.[175 182 187] In qualitative studies the focus was on rapid methods for data analysis,[175 182–186 190 191] including three studies which compared rapid and standard analysis (see Studies comparing a rapid with a non-rapid method, below).[183 185 186] Here, as in the section above "Increasing rapidity by doing less", there was some overlap with the use of technological innovation to increase rapidity (see section below: "Using alternative technology or data source").[181] This is particularly the case with the use of web-based or virtual technology in specific stages

**Table 3** Adaptation of a discrete stage of a non-rapid evaluation to make this a rapid part of the wider study

| Study stage targeted | Rapid methods | Examples |
|---|---|---|
| Priority setting | To improve rapidity of research question prioritisation (applicable to multiple designs). Use of virtual meetings in some instances. | Gray et al, Newlands et al, Cowan et al, Dudovitz et al[133 163 193 194] |
| Ethics approval | Intended to reduce time to obtain ethical approval by addressing requirements for multiple approvals (applicable to multiple designs). | Khan et al[164] |
| Recruitment and participation | Reducing time taken to identify participants and complete recruitment through methods to prespecify or identify potential participants, (primarily in RCTs) including through using participants from previous studies (in surveys); use of near real-time data from test-and-trace to identify potential participants in time-sensitive trial. | Bower et al, Brown et al, Deane et al, Corley et al, Xiang et al, Wesolowski et al, Cake et al[165–167 172 173 195 196] |
| | Using multiple methods for recruitment alongside rapid analysis of success for each method. Use of websites or social media to identify and recruit participants for surveys or interview studies (conduct of interview studies noted below). | Bjornson-Benson et al, Attal et al, Daviskiba et al, Elsawy et al, Hammond et al, Khubchandani et al, Talati et al, Petrella et al[27 145–150 154] |
| Rapid bilingual appraisal | Reducing barriers to inclusion in rapid evaluation; applicable to multiple study designs where language is a potential barrier. | Whelan[168] |
| Assessment of barriers to process | Tool for rapid assessment of barriers to process; assessed within an RCT. | Alolod et al, Traino et al[169–171] |
| Structure of treatment groups/assignment | Proposal for flexible structuring of treatment assignment and use of single control group for multiple potential vaccine candidates in multistage RCT. Implementation of platform trials for multiple intervention assessment. | Moodie et al, Douglas et al, Griffiths et al, Murray et al, Short et al, RECOVERY[162 174 176–180 243] |
| Nesting of additional studies | Continuous rapid data analysis in realist study embedded in standard RCT. | Hensel et al[175] |
| Data collection | Bespoke reporting platform; cohort study protocol to rapidly assess vaccines as well as epidemiology of disease. | Simpson et al[181] |
| | Data tracking and rapid continuous feedback to participants (My Own Health Report). Designed to provide continuous collection and analysis of data in a pragmatic RCT. | Glasgow et al[182] |
| | Use of multiple publicly available data sets to evaluate vaccine impact. | Li et al[187] |
| | Use of online, web-based methods to interview individual participants as alternative or optional method or to conduct focus groups. | Fisher et al, Keniston et al[151 153] |
| Data analysis | Rapid analysis of data with or without other rapid stages; used in qualitative (comparative) and mixed methods studies; embedded realist study; cohort study; focus group. Specific methods include Rapid Identification of Themes from Audio recordings. Other methods noted include measures to improve rigour (eg, triangulation, reflexivity, peer examination). | Taylor et al, Burgess-Allen and Owen-Smith, Glasgow et al, Hensel et al, Nevedal et al, Vázquez et al, Rolf et al, Lewinski et al, Tan et al, Elwy et al, Gale et al[175 182–186 188–192] |

of studies. We noted increasing use (from a low base) of methods for rapid analysis of qualitative data in the context of the COVID-19 pandemic.[175 182–185 190 191 197]

### Using alternative technology or data source

The use of different or newer technology to rapidly acquire and/or analyse routine data was another approach to making evaluation more rapid (table 4). As we have noted there is some overlap between this and the classifications above, but our synthesis considered this separately as it represents a technological streamlining of an existing process rather than an adaptation of the research design.[58 138] Perhaps unsurprisingly this is an area in which we see the emergence of substantial numbers of studies and new adaptations of methods with the response to the COVID-19 pandemic.

The 28 studies discussed here involved using technology to automate or otherwise try to improve rapidity of one or more of data collection, collation or analysis.[58 129 138 145–151 153 154 173 181 197–209] Studies which involved developing, acquiring, sharing or using existing data sets to enable modelling of effectiveness were also included.[29 129 173 200 203 206–210] Seven studies were undertaken in the context of monitoring of uptake and safety profiles of vaccinations even prior to COVID-19. For study details see online supplemental appendix 2 tables 2 and 3.

### Studies comparing a rapid with a non-rapid method

We found only four studies which compared a rapid evaluation method with a non-rapid method, and we did not find any comparisons of different types of rapid evaluations. Three of the studies focused on comparing rapid versus traditional analysis of qualitative data.

One study compared the use of rapid evaluation with standard evaluation of an enhanced health programme in care homes. The study was described as assessing rapid evaluation but the rapid evaluation itself took 2.5 years.[143] Nevertheless, this was the only explicit comparison we

**Table 4** Use of an alternative technology or data source

| Study authors' suggested rapid innovation | Research design | Impact/intention of method | Example(s) |
|---|---|---|---|
| New digital systems | Registry study Surveys | Introduction of new computerised systems to assess vaccination coverage of routine immunisation programmes; computerised child registers also became available. Computer-assisted telephone surveys. Authors stated that it enabled more timely feedback to coordinators of district vaccination programmes and rapid identification of random samples for interview. Online surveys; use of wholly online surveys hosted on webpages or social media to recruit participants and conduct survey. Computer-based rapid recruitment. Computer-based rapid data-collection/analysis. | Williams *et al*, Wichmann *et al*, Begg *et al*, Attal *et al*, Daviskiba *et al*, Elsawy *et al*, Hammond *et al*, Khubchandani *et al*, Talati *et al*, Petrella *et al*[58 138 145–150 154 198] |
| Use of videoconferencing technology | Assessments or evaluations using interviews or focus groups | Videoconference platforms used to enable interviews or focus groups; in some cases multiple sessions were conducted simultaneously. | Fisher *et al*, Keniston *et al*[151 153] |
| Using bespoke data platform | Cohort | Nationally representative sample assessed using linking of multiple data sources from clinical practice, laboratory results and death certification. Designed to assess pandemic influenza reporting and vaccination impact; plan to be implemented for future influenza pandemic. | Simpson *et al*[181] |
| Secure web-based tool for pharmacovigilance | Cohort | Secure web-based tool for small number of set fields focused on single medications in hospice context using rapid prospective reporting at agreed time points to allow rapid aggregation of data on small numbers of participants from multiple centres. | Currow *et al*[199] |
| Automation of data collection | Registry study | Use of hospital routine data collection to populate registry with validation to analyse costs for two different methods of triage of designated patient group. | Noack *et al*, Price *et al*[129 200] |
| Automation of data processing | Registry study | Use of regional/organisational vaccine information system to rapidly assess influenza vaccine campaign safety and coverage. | Alguacil-Ramos *et al*, CDC[201 204] |
| Use of existing data sets | Registry-based studies | Use of data from (eg,) an insurance database or hospital data registry to populate a comparison of a specified adverse event or to analyse costs. Use of real time data within an organisation to create a Learning Healthcare System. Using a national database for near real-time recruitment in time-sensitive treatment trial. Methods of data-sharing or promotion of data-sharing, including creation of publicly accessible rapidly updated data repositories. | Gagne *et al*, Noack *et al*, Agaku *et al*, Cake *et al*, Carrillo *et al*, Dambha-Miller *et al*, Harrison *et al*, Talwai *et al*, Saunders nd Gkousis, Price *et al*[129 173 200 202 205 209 210] |
| Use of technology to generate data sets | Qualitative study | Use of orchestrated Twitter chat to generate data set for qualitative analysis. | Shimkhada *et al*[197] |
| Simulation of impact of vaccination using existing healthcare data sets | Simulation study | Use of established data sets to run simulations on how vaccine coverage and population characteristics impact ability to detect safety signals for influenza vaccinations. The authors recommended this be used only as a method to detect signals and generate hypotheses. | Dodd *et al*[203] |

found. This study assessed the cost consequences of implementing the programme and compared the use of shorter and longer-term interrupted time-series (ITS) analyses, using shorter and longer time-series of aggregated data.[143] The authors found that the longer-term analyses yielded a different interpretation, an improved

model fit and greater precision; and they considered that the shorter-term analyses produced results likely to be misleading. The time period required for acceptable internally valid ITS based studies is contested more generally.[211–214]

A second study compared rapid with usual qualitative analysis in an evaluation of a home birth service in the UK.[183] The study authors reported on the total time taken in the data-management process as well as the analysis component. The agreement between the findings and recommendations of the two processes was also considered. The authors concluded that rapid analysis delivered a modest time saving which may have been greater if there had been greater concordance between the characteristics of the two research teams. They also reported substantial overlap in findings between the two analysis approaches, although their recommendations diverged more. The two analyses were undertaken by different research teams which had differing levels of contextual experience, different stakeholder audiences and differing opportunities for analysis discussion and this was identified as a limitation in the study, together with the fact that rapid analysis was a novel approach for the researchers involved. The study authors concluded that rapid analysis may potentially deliver valid findings while taking less time but recommended further comparisons using additional data sets with more comparable research teams.

Two studies compared rapid with standard qualitative analysis within the US Veterans Health Administration.[185 186] One study looked at an evaluation of a programme which seeks to identify and diffuse evidence-based interventions,[185] while the other evaluated a specific evidence-based strategy in opioid prescribing.[186] In each case authors used the Consolidated Framework for Implementation Research and compared the use of a rapid deductive approach to analysis using notes and audio recordings to a traditional deductive approach using transcripts.

For the wider programme authors reported that the final summaries produced by the two approaches were 'quite similar'; they judged that both approaches allowed them to identify and describe the factors influencing implementation, which was the goal of their evaluation, but that the rapid approach allowed formal results to be generated and shared more quickly.[185] In the evaluation of the opioid prescribing programme the authors reported that findings from the two analyses were consistent.[186]

## DISCUSSION
### Summary
This scoping review identified a lack of clarity in approaches described as 'rapid evaluation' in primary research. The approaches to rapid evaluation which we identified could be grouped into four broad categories (1) studies using specific methodologies designed for rapid evaluation; (2) studies which carried out less evaluation or less intensive evaluation ; (3) studies which

used rapid methods only for a particular stage of evaluation; and (4) studies which used technology to increase rapidity. There was very little comparative research on the impact of using rapid rather than non-rapid evaluation.[143 183 185 186]

It is notable that we did not identify many rapid evaluation methodologies clearly designed to assess the comparative effectiveness of innovations. This was reflected in the fact that the recent systematic review which focused on rapid evaluation methodologies more narrowly included only 12 studies despite considering studies from LMIC as well as high-income countries.[9] Studies using bespoke, rapid methodologies used qualitative or mixed methods and mainly assessed aspects of user experience and acceptability along with associated barriers and facilitators, and the implementation of innovations in particular contexts. This may reflect the types of questions which are considered suitable for rapid evaluation and the impetus to develop rapid evaluation methods in these areas of research, but it may also reflect conceptions of what evaluation is possible within a limited time frame.

### Impact of COVID-19 pandemic
Our initial search was conducted in February 2020 and our update search in September 2022, meaning that we were able to identify the impact of the COVID-19 pandemic on the published literature to date. Among the 162 included studies 90 were identified from the update search in September 2022. The fact that over half of all included studies were indexed since the onset of the pandemic indicates a substantially increased level of rapid research publication.

This increase in publication numbers has coincided with the accelerated development of some approaches to rapid evaluation previously identified as proposals or isolated examples of methods. The use of 'platform' designs for flexible participant allocation within RCTs is exemplified by the RECOVERY trial[180] but we also found recent studies using this approach in non-COVID contexts such as oncology. The use of rapid analysis methods within qualitative studies also showed more examples, methodological developments and assessment of the impact of using rapid compared with conventional methods. The impact of COVID-19 is also clear in participation and recruitment phases of studies. We identified an example of linking of almost real-time data sets to facilitate recruitment in a trial of a time-sensitive intervention. This study used data from the UK National Health Service (NHS) test-and-trace programme to identify high risk people who had recently recorded a positive test for COVID-19 in the community for recruitment to a treatment trial.[173] We also see the first examples of web-based methods used to rapidly generate data for evaluation.[197]

We also identified papers relating to infrastructure designed to increase research rapidity and coordination. This involved the creation of data repositories which can incorporate real world data from different sources, thereby supporting further studies which use existing

data sets to enable rapid evaluation, and triangulation of data from different sources.[206–208 215] We also identified methodological work to increase the comparability and useability of data entering such repositories.[205] The importance of infrastructure and of leveraging existing resources and relationships in supporting rapid evaluation more widely was the subject of a number of commentaries identified by our searches but not eligible for inclusion.[216–219]

However, most of the newly identified studies represented one of two approaches to rapid evaluation. There was an increase in the use of established rapid evaluation methodologies such as rapid appraisal. Many of these recent studies also involved an adaptation of the processes used for remote working approaches with online recruitment and videoconferencing incorporated into the established methodologies. We also found multiple studies which used remote working approaches (to all or part of a study) as a less time intensive and technological approach to make a study rapid. Examples of this were use of online surveys, online recruitment or videoconferencing software for interviews or focus groups. The adoption of these methods within and outside bespoke methodologies may have been driven by necessity but is likely to represent a permanent change in the rapid evaluation toolkit. The impact using remote methods on research has not yet been systematically explored.

### Limitations of the review

We registered our protocol at an early stage of the review process and specified in it that we would adopt a flexible and iterative approach to the review. We used this flexibility to narrow the scoping review we conducted in several ways in view of the very large number of identified papers. We addressed only the two questions we considered central to mapping the evidence around rapid evaluation and limited the eligible studies to those concerned with primary evaluation research in high-income countries, excluding regulatory processes as well as rapid reviews. We acknowledge that there is the potential for these decisions to have been driven by the nature of the identified literature, but we prespecified this flexible approach to produce a review that would be relevant to the context of the NHS in England, while remaining comprehensive enough to capture relevant research methods. We have documented this in our methods and results to ensure transparency about the review processes.

The ongoing work by others in documenting and developing methods for rapid reviews means that we are confident that our decision to focus on primary research, where there is much less evidence synthesis or guidance is justified. Our decision to exclude reports from a regulatory context has the potential to exclude relevant studies, particularly given the pivotal role of the UK National Institute for Health and Care Excellence (NICE) in developing approaches to such evaluation. However, after coding the identified studies it was apparent that these rapid methods related primarily to evidence synthesis.

Future research could usefully explore the impact of a rapid or truncated regulatory process on the types of evidence considered sufficient for approvals, and the implications of this for primary evaluation.

A related limitation concerns the decision to limit our inclusion criteria to reports from high-income countries, as we may have omitted methods for rapid evaluation which might be transferable to the UK context. First, we identified the use of lot-quality surveys (a method of using small sample sizes to determine if areas have achieved a specified target) as a method of doing research with fewer participants; this was only represented in studies from LMIC.[161 220] We also found an example of a method of increasing rapidity by making part of a study more rapid which was only identified in LMIC contexts but may be particularly relevant to disadvantaged or marginalised participants in the UK. Six studies used preparatory methods to ensure informed consent so that enrolment for subsequent studies could be undertaken rapidly because necessary groundwork had been undertaken.[168 221–225] We note that some of our excluded studies from LMIC described their methods as 'rapid' but with limited information on what promoted rapidity. Our initial assessment suggested that evaluations completed within a period of weeks, or a few months was what qualified them as rapid in the eyes of the authors. It is possible that a systematic review with wider inclusion criteria and the resources to contact authors may have identified specific methods which would have relevance to UK settings, but which are not represented in either our included studies or the excluded studies summarised here.

Due to the high volume of identified literature at the initial and second stages of the review process we did not use duplicate screening and coding by two independent researchers for all studies. We acknowledge that this may have increased the risk of error or bias in the review processes. However, we screened and coded subsets of the records at each stage in duplicate and found very high levels of agreement, so we do not believe that the impact is likely to be substantial. This is particularly the case as we consulted a second reviewer in cases of uncertainty for other records, both in screening and in coding, meaning that consensus was used where judgements were difficult.

Our review complements a recently published systematic review of rapid evaluations.[9] Our review had a wider methodological scope and included a correspondingly larger number of studies (162 studies compared with 12), but identified some similar themes, such as variable labelling of rapid evaluations, and similar strategies to achieve rapidity. Importantly, the previous review highlighted the importance of the 'trustworthiness' of rapid designs, while we were able to explore the limited but critical evidence base comparing rapid and non-rapid methods. We have included both this systematic review and a subset of the primary studies which it included, however this corresponds to only eight primary studies across different sections of our results, so we believe the impact of potential double-counting to be minimal.[9]

Because this is a scoping review we did not conduct a quality assessment of the identified studies; we therefore document the nature of the research available rather than assessing its reliability.[23 24] We do not attempt to evaluate the success or otherwise of methods proposed as enabling rapid evaluation but have accepted study authors' own assessments of the usefulness or success of their methods. A further limitation of this review is that we were unable to identify (and therefore include) studies that may have carried out work which represents rapid evaluation but have failed to describe it as such (eg, retrospective quantitative analyses). This might bias the mapping of studies identified if particular types of studies exhibit differential propensity to report themselves as rapid.

Conversely, we identified some evaluations which the study authors described as rapid, but which might not commonly be considered as rapid, except in comparison to an even more detailed and lengthy standard process. We explicitly excluded pilot studies (often RCTs) because these were considered to be assessing feasibility rather than undertaking evaluation.[226] This may have contributed to a skewing of the literature identified towards non-quantitative methods, but the pilot studies we did identify did not label themselves as being rapid studies.[227–229] We are aware that we may therefore have omitted some examples of studies which would meet some definitions of rapid. However, these limitations would not explain the relative paucity of studies comparing rapid to alternative methods in the identified literature, and this appears to be a genuine evidence gap.

## Interpretation of the studies

The drive towards rapid evaluation raises tensions for those commissioning and delivering research. Evaluation studies aim to achieve rigour (broadly, internal and external validity, or their qualitative equivalents) and scope (answering a range of questions, including those relating to access, effectiveness, cost effectiveness, acceptability, equity and implementation). In most circumstances these must be achieved under constraints of cost and time. The greater the scope of the project, and the greater the protection against validity threats, the higher is the cost and time. These known trade-offs are being explored by organisations such as BRACE and RSET.[230]

Occasionally, researchers can trade time against cost. For example, national recruitment drives (such as those under COVID-19 in trials such as RECOVERY) have maintained high levels of rigour under time pressure by deploying significant resources.[180] Some of the factors delaying evaluation delivery can be managed through additional resources. However, some, such as long-term follow-up, cannot be managed in that way. In regulatory contexts these trade-offs may be addressed through mechanisms of post-marketing surveillance,[231] but this only applies to a subset of innovations such as pharmaceuticals or medical devices.

Where studies cannot be accelerated through additional resources, or where costs are constrained, rapid evaluation essentially involves trading off rigour or scope against time. Some of those trade-offs can be profound. For example, a study of an innovation might eschew questions of comparative effectiveness and cost-effectiveness entirely and restrict the scope of the study to issues of staff acceptability and implementation. This seemed to be the approach adopted in some of the studies identified in the review. Quantitative rapid evaluations are more likely to evaluate impacts of innovations on short-term indicators (such as processes of care) which are often proxies for outcomes where longer-term follow-up is required. Therefore, the limitations of such approaches are likely to depend on the strength and consistency of the relationship between the proxy and the ideal longer-term outcome. Some decision-making frameworks (such as the NICE digital framework) specifically identify some interventions where such trade-offs are legitimate.[232] Contextual pressures (such as the pandemic) may force trade-offs to ensure timely decisions, but they have also been the impetus for improved infrastructure which may support ongoing rapid research.[215] We noted concerns that ethics and stakeholder engagement should not be impacted negatively by increased rapidity.[233–235]

Other trade-offs will be more marginal. For example, a decision maker might accept a smaller sample (and hence a lower level of quantitative precision) or more selective recruitment (at some cost to external validity). It is also the case that reduced statistical power related to using shorter post-intervention time periods might be at least partly offset by using a comparative design (such as difference-in-differences) rather than interrupted time series, since estimates from comparative designs are based on more information than non-comparative approaches. Such trade-offs impact rigour and scope in a more limited way that might be more acceptable. Larger trade-offs will be potentially easier to assess in terms of their impact—for example, eschewing questions of comparative effectiveness will, or should, restrict what can be claimed about an innovation. Some trade-offs are more difficult to judge in terms of their impact.

The role of rapid evaluation methodology is to reduce impacts on rigour and scope as far as possible, while maintaining the benefits in terms of time. We identified several examples of these in the literature, although there was limited comparative work on their success.[143 183] Since conducting prospective comparisons of rapid and non-rapid evaluation methods is time-consuming and resource intensive, it may be possible to use retrospective analysis of existing data to explore the impact of using shorter analysis periods. For example, it would be possible to conduct a sensitivity analysis of truncated sets of time points in an interrupted time series, and to compare the results to the full analysis. Performing such analyses on several sets of data from different studies would allow assessment of whether there was a consistent impact on results but would be possible with only a limited set of study designs. In this, studies based on regulatory

decision-making, excluded from this review, may provide useful design features.

## Future research

There is a need for clarity and consistency in terms of what constitutes rapid evaluation and, in particular, what differentiates rapid from non-rapid evaluation. Consensus on definitions and more consistent labelling will be important in allowing ongoing assessment of this developing field. We suggest that better description would involve clarity about what aspects of the research process were conducted more rapidly, when this occurred in the research process and what the potential impacts could be.

The development of specific rapid methodologies and technologies is needed to provide a toolkit for research teams to meet the challenges of rapid evaluation.

The development of new, rapid methodologies needs to be matched by comparative work on their advantages and disadvantages, such as the impact on uncertainty for decision-makers. Such comparative work will be challenging, adding cost and complexity in a context where commissioners and researchers are seeking to reduce both and will require innovative methodologies, such as testing rapid methods nested within 'standard' projects, for example[175 182] to maximise efficiency.

**Author affiliations**
¹Division of Nursing, Midwifery & Social Work; School of Health Sciences; Faculty of Biology Medicine and Health, The University of Manchester, Manchester, UK
²Manchester Academic Health Science Centre, Research and Innovation Division, Manchester University Foundation NHS Trust, Manchester, UK
³Centre for Primary Care and Health Services Research; School of Health Sciences; Faculty of Biology, Medicine & Health, University of Manchester, Manchester, UK
⁴Division of Health Research, Lancaster University, Lancaster, UK

**Acknowledgements** The authors are grateful to Dr Su Golder for design and implementation of the search strategy and to Sophie Bishop for the search update.

**Contributors** GN and TDM carried out the review process with input from JCD, PB, PW and NC. All authors contributed to interpretation of the data. GN wrote the first draft of the manuscript text. JCD, TDM, PB, PW and NC reviewed and substantially edited and revised the manuscript. GN is the guarantor of the manuscript.

**Funding** This research was funded by the National Institute for Health and Care Research Applied Research Collaboration Greater Manchester. The views expressed in this publication are those of the author(s) and not necessarily those of the National Institute for Health and Care Research or the Department of Health and Social Care.

**Competing interests** None declared.

**Patient and public involvement** Patients and/or the public were not involved in the design, or conduct, or reporting, or dissemination plans of this research.

**Patient consent for publication** Not applicable.

**Ethics approval** Not applicable.

**Provenance and peer review** Not commissioned; externally peer reviewed.

**Data availability statement** All data relevant to the included studies are included in the article or uploaded as supplementary information. A list of studies excluded at full text is available on reasonable request. This is a scoping review of previously published studies, so all the data used in the review is already in the public domain. Full lists of all studies considered after initial screening are available from the corresponding author on reasonable request.

**ORCID iDs**
Gill Norman http://orcid.org/0000-0002-3972-5733
Thomas Mason http://orcid.org/0000-0003-3135-0364
Jo C Dumville http://orcid.org/0000-0002-6546-3685
Peter Bower http://orcid.org/0000-0001-9558-3349
Paul Wilson http://orcid.org/0000-0002-2657-5780
Nicky Cullum http://orcid.org/0000-0003-2631-123X

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
