## [Reviewer comments · BMJ Open]

ARTICLE DETAILS

TITLE (PROVISIONAL)	Approaches to enabling rapid evaluation of innovations in health and social care: a scoping review of evidence from high income countries
AUTHORS	Norman, Gill; Mason, Thomas; Dumville, Jo C.; Bower, Peter; Wilson, Paul; Cullum, Nicky

VERSION 1 – REVIEW

REVIEWER	Corbett, Mark University of York, CRD
REVIEW RETURNED	26-Jul-2022

GENERAL COMMENTS	1. Searches The main limitation of this scoping review is how long ago the literature searches were made. The review is out of date to the extent that research relating to the Covid pandemic could not have been included (the searches were performed in February 2020). A review update is needed given both the review being over two years old and the likely importance of the use of rapid methods in COVID-19 research. Discussion should also be made of a related systematic review published last year: Rapid, Responsive, and Relevant?: A Systematic Review of Rapid Evaluations in Health Care by Vindrola-Padros et al. (American Jnl of Evaluation). 2. Eligibility criteria Rapid reviews were ultimately excluded from the review - “we subsequently narrowed the scope of our search to exclude rapid reviews” (p6, line 19-20). Since identification of rapid reviews formed a key part of the search strategy, I think a clearly stated rationale for excluding rapid reviews is needed. I’m also a little confused by the meaning of “...exclusion of studies which looked at evaluation in a regulatory context because these were more closely related to review methodology”. Would, for example, a regulatory evaluation which compared approvals using surrogate outcomes (rapid) versus real clinical outcomes (non-rapid) to gain regulatory approval be excluded? That example is hypothetical – but I’m wondering whether excluding regulatory evaluations might have resulted in some methods being missed? Perhaps listing these excluded studies would be useful. Also, isn’t reference 95 (Boucaud-Maitrels) an evaluation in a regulatory context? 3. Protocol The study protocol states that the review will include previous systematic or scoping reviews of rapid evaluations or types of
---

	evaluations and that where an appropriate review is identified they will not also include the primary studies which are included in it. I did not see reference to this criterion in the manuscript - please could the authors clarify whether these reviews were eligible (and why the criterion was dropped, if that was the case). Some of the review questions (in the protocol) do not appear to have been answered in the manuscript e.g. What centres of expertise in rapid evaluation are there? What guidance exists for those undertaking and evaluating rapid evaluations in health and social care? If these questions are answered in the manuscript could this be made clearer (using sub-headings)? Alternatively, if the scope of this review has narrowed since the protocol was written, this should also be clearly explained. 4. Other Please define LMIC at first use.
--	--

REVIEWER	Pelletier, Chelsea University of Northern British Columbia
REVIEW RETURNED	29-Aug-2022

GENERAL COMMENTS	This is an interesting paper exploring rapid evaluations conducted in health or social care settings. The methods are rigorous and strengthened by preregistration of the protocol. I have noted some discrepancies between the protocol and manuscript that I think should be addressed.  1. The background is quite brief and does not really establish a strong justification for the study or a review of relevant literature. Currently, it reads more like a list of definitions – innovations, evaluation, rapid evaluation – similar to the bulleted list provided in the protocol. While I think it is important to provide definitions for the terms adopted for this review, I wonder if they may fit better in a table, and the background used to introduce the need for your review. For example, have there been other reviews conducted on rapid (or other) evaluation methods? Why do we need to understand the methods? Page 3, Lines 36-39 provide a small amount of this but are two fairly vague sentences. 2. There are several research questions included in the protocol that do not appear in the paper. Please explain. 3. The inclusion criteria for ‘study design’ in the protocol included “previous systematic or scoping reviews of rapid evaluations or types of evaluations...”. However, in the paper (table 1, study design) you state “we did not include rapid reviews or papers relating to methods for rapid reviews”. Please clarify the parameters for including/excluding secondary data sources. 4. Search (page 5, line 7) indicates the search was conducted in February 2020. Given this search is now more than 2 years old, was there any effort to provide a more up-to-date search? Particularly given the plethora of COVID-related work published in the past 2 years (which is used as justification for this review). 5. Screening (page 7, line 12) and data extraction (page 7, lines 28-29). Please describe how conflicts between reviewers were managed (uncertainty). Eg, consensus through discussion, third reviewer. The limitations (page 20, line 42) indicate a second reviewer was consulted but this should be made more clear in the methods. Was inter-rater agreement calculated to support the statement of “very high levels of agreement”? 6. Results, line 19. Indicates 420 records were assessed in full text.
---

	The abstract (page 2, line 25/26) indicates 352 were evaluated in full text. 7. More information about the synthesis and analysis processes is necessary. How were the themes identified (page 8, line 8/9)? Are the themes the categories described elsewhere in the results section? 8. I like how the results section has been structured based on categories of rapid review approaches.
--	---

VERSION 1 – AUTHOR RESPONSE

Reviewer		
1		
1	Searches The main limitation of this scoping review is how long ago the literature searches were made. The review is out of date to the extent that research relating to the Covid pandemic could not have been included (the searches were performed in February 2020). A review update is needed given both the review being over two years old and the likely importance of the use of rapid methods in COVID-19 research. Discussion should also be made of a related systematic review published last year: Rapid, Responsive, and Relevant?: A Systematic Review of Rapid Evaluations in Health Care by Vindrola-Padros et al. (American Jnl of Evaluation).	Searches We have updated the search in September 2022 and have revised the review accordingly. Because the original and update searches capture the impact of the covid-19 pandemic on the published literature we have used this as an opportunity to explore this briefly in the discussion section. Relevant review We have now included and briefly discussed the highlighted systematic review. In our discussion we have also commented on the findings of the systematic review and considered its relationship to our own scoping review; this is interwoven into the discussion and can be found in the summary and in the limitations of the review sections. We have specifically addressed the issue of overlap between the two reviews in terms of breadth and number of included studies. The overlap in included studies is limited as we had much wider methodological inclusion criteria while the systematic review included LMIC studies as well as those from high income counties.
2	Eligibility criteria Rapid reviews were ultimately excluded from the review - “we subsequently narrowed the scope of our search to exclude rapid reviews” (p6, line 19-20). Since identification of rapid reviews formed a key part of the search strategy, I think a clearly stated rationale for excluding rapid reviews is needed. I’m also a little confused by the meaning of “...exclusion of studies which looked at evaluation in a regulatory context because these were more closely related to review methodology”. Would, for example, a regulatory evaluation which compared approvals using surrogate outcomes (rapid) versus real clinical outcomes (non-	Rapid reviews This is an excellent point, and we are happy to clarify the reasons for this decision which was made in view of the high volume of literature and rapidly developments in in rapid review methods. Since our main interest was primary research, we focused our review on this. We have added the following text to explain our decision and to direct readers to work which deals with rapid reviews. “We subsequently narrowed the scope of our search to exclude rapid reviews (Table 1) as it became clear that the volume of rapid reviews and related methodological papers would require a separate piece of work. Given that considerable work on rapid review methodology is being

	rapid) to gain regulatory approval be excluded? That example is hypothetical – but I’m wondering whether excluding regulatory evaluations might have resulted in some methods being missed? Perhaps listing these excluded studies would be useful. Also, isn’t reference 95 (Boucaud-Maitrels) an evaluation in a regulatory context?	undertaken (18-21) we took the decision to focus on primary research.” We refer the interested reader to appropriate references for this methodological work. Our update search in September 2022 did not include this facet. Regulatory studies The regulatory studies we looked at were primarily methods of accelerating approval of pharmaceutical interventions. As such they were using shortened evidence review procedures which was why we ultimately felt that they did not usefully contribute to this review, having made the decision to focus on primary research. We have edited the text to make this clearer. It now reads “We also excluded studies which looked at evaluation in a regulatory context because these represented accelerated evidence review processes and were therefore considered to represent review methodology rather than primary evaluation.” However, the reviewer raises an important point about the implications of an accelerated process, and we have added what we feel is a useful paragraph to the “limitations” section of the discussion in response; we’re grateful for the opportunity to explore this aspect. Ref 95 Boucaud-Maitrels We are grateful to the reviewer for identifying this mistake. Reference 95 in the supplementary tables is Boucaud-Maitrels 2016 which is not included as a reference in the main text of the review because it was excluded when we made the decision to exclude regulatory studies. It was not removed from the supplementary tables in error. We have now removed it and have cross-checked these tables for other such errors.
3	Protocol The study protocol states that the review will include previous systematic or scoping reviews of rapid evaluations or types of evaluations and that where an appropriate review is identified they will not also include the primary studies which are included in it. I did not see reference to this criterion in the manuscript - please could the authors clarify whether these reviews were eligible (and why the criterion was dropped, if that was the case). Some of the review questions (in the protocol) do not appear to have been answered in the manuscript e.g. What	Previous systematic or scoping reviews Existing systematic or scoping reviews are eligible. We did not identify any eligible systematic or scoping reviews of rapid evaluations (reviews of rapid reviews and rapid reviews were excluded) in our initial work. The phrasing of the eligibility criteria “we planned to summarise this” was intended to convey this. However, in our update search we have identified one such review – that identified by the reviewer in their comment (1) above. Given that this was identified in an update we have taken the decision to include the review and to note which of our included primary studies are included in

	centres of expertise in rapid evaluation are there? What guidance exists for those undertaking and evaluating rapid evaluations in health and social care? If these questions are answered in the manuscript could this be made clearer (using sub-headings)? Alternatively, if the scope of this review has narrowed since the protocol was written, this should also be clearly explained.	it. We have identified this as a limitation in the discussion section. We have therefore edited our inclusion criterion to reflect this amendment. Review questions We have clarified that we have addressed fewer questions in the review than were originally planned for in the protocol. “In particular we have narrowed the scope of the review and focused on addressing a subset of the original questions posed. We have done this in order to engage fully with the questions we considered most central to our objectives (see above).” We have also added this to the limitations section of the discussion.
4	Other Please define LMIC at first use.	We have provided the definition low and middle income countries at first use.
Reviewer 2	This is an interesting paper exploring rapid evaluations conducted in health or social care settings. The methods are rigorous and strengthened by preregistration of the protocol. I have noted some discrepancies between the protocol and manuscript that I think should be addressed.	Thank you. We have sought to address the specific comments below.
1	1. The background is quite brief and does not really establish a strong justification for the study or a review of relevant literature. Currently, it reads more like a list of definitions – innovations, evaluation, rapid evaluation – similar to the bulleted list provided in the protocol. While I think it is important to provide definitions for the terms adopted for this review, I wonder if they may fit better in a table, and the background used to introduce the need for your review. For example, have there been other reviews conducted on rapid (or other) evaluation methods? Why do we need to understand the methods? Page 3, Lines 36-39 provide a small amount of this but are two fairly vague sentences.	Thank you for this. While we have not used a table for the definitions, we have moved them into the Methods section and have provided a longer background section instead. We feel this improves the structure of the review and we appreciate the useful suggestion.
2	2. There are several research questions included in the protocol that do not appear in the paper. Please explain.	Please see our response to reviewer 1 above – we are grateful to both reviewers for raising this and have clarified in the methods and added further text to the limitations section of our discussion.
3	3. The inclusion criteria for ‘study design’ in the protocol included “previous systematic or scoping reviews of rapid evaluations or types of evaluations...”. However, in the paper (table 1, study design) you state “we did not include rapid reviews or papers relating to methods for rapid reviews”. Please clarify the parameters for including/excluding secondary data sources.	We included systematic or scoping reviews of primary rapid evaluations. We did not include rapid reviews of evaluations (i.e. where it was the synthesis process that represented the rapid element. We have clarified this in the text, thank you for highlighting the ambiguity.
4	4. Search (page 5, line 7) indicates the search was conducted in February 2020.	We have updated the search in September 2022 and have revised the review

	Given this search is now more than 2 years old, was there any effort to provide a more up-to-date search? Particularly given the plethora of COVID-related work published in the past 2 years (which is used as justification for this review).	accordingly. Because the original and update searches capture the impact of the covid-19 pandemic on the published literature we have used this as an opportunity to explore this briefly in the discussion section.
5	5. Screening (page 7, line 12) and data extraction (page 7, lines 28-29). Please describe how conflicts between reviewers were managed (uncertainty). Eg, consensus through discussion, third reviewer. The limitations (page 20, line 42) indicate a second reviewer was consulted but this should be made more clear in the methods. Was inter-rater agreement calculated to support the statement of “very high levels of agreement”?	We did not formally calculate inter-rater agreement but are happy to clarify the basis for our statement that we had very high levels of agreement. As we noted in the section “screening” we identified only a single record excluded at first stage screening which represented a subsequently included record. We have added more information on this however – our double screened sample identified 3 records which required full-text screening and one of these was subsequently included.
6	6. Results, line 19. Indicates 420 records were assessed in full text. The abstract (page 2, line 25/26) indicates 352 were evaluated in full text.	We apologise for the confusion here which occurred with the decision to exclude papers in LMIC following full-text evaluation. We have edited both the abstract and the main results section to accurately reflect the results of the review, including the updated search from September 2022.
7	7. More information about the synthesis and analysis processes is necessary. How were the themes identified (page 8, line 8/9)? Are the themes the categories described elsewhere in the results section?	Yes, these themes are the categories described in the results section. We have clarified this and have used the word approaches in order to avoid confusion. We have also provided a short description of how the theme/categories were identified.
8	I like how the results section has been structured based on categories of rapid review approaches.	Thank you – no response required

VERSION 2 – REVIEW

REVIEWER	Corbett, Mark University of York, CRD
REVIEW RETURNED	30-Nov-2022

GENERAL COMMENTS	I am satisfied that all the points raised have been appropriately addressed in the revised version, although it would benefit from being re-read for minor typos and missing words.
---

REVIEWER	Pelletier, Chelsea University of Northern British Columbia
REVIEW RETURNED	23-Nov-2022

GENERAL COMMENTS	Thank you for addressing all of my concerns.
--